# Study on High-Temperature Mechanical Properties of Fe–Mn–C–Al TWIP/TRIP Steel

**Guangkai Yang [1,2], Changling Zhuang [1,2,\*], Changrong Li [1,2], Fangjie Lan [1,2] and Hanjie Yao [1,2]**

1   School of Materials and Metallurgy, Guizhou University, Guiyang 550025, China; 18085037568@163.com (G.Y.); crli@gzu.edu.cn (C.L.); fangjielan@163.com (F.L.); yhj1394542756@163.com (H.Y.)
2   Guizhou Key Laboratory of Metallurgical Engineering and Process Energy Conservation, Guiyang 550025, China
\*   Correspondence: clzhuang@gzu.edu.cn; Tel.: +86-186-0851-4532

**Abstract:** In this study, high-temperature tensile tests were carried out on a Gleeble-3500 thermal simulator under a strain rate of $\varepsilon = 1 \times 10^{-3} \text{ s}^{-1}$ in the temperature range of 600–1310 °C. The hot deformation process of Fe–15.3Mn–0.58C–2.3Al TWIP/TRIP at different temperatures was studied. In the whole tested temperature range, the reduction of area ranged from 47.3 to 89.4% and reached the maximum value of 89.4% at 1275 °C. Assuming that 60% reduction of area is relative ductility trough, the high-temperature ductility trough was from 1275 °C to the melting point temperature, the medium-temperature ductility trough was 1000–1250 °C, and the low-temperature ductility trough was around 600 °C. The phase transformation process of the steel was analyzed by Thermo-Calc thermodynamics software. It was found that ferrite transformation occurred at 646 °C, and the austenite was softened by a small amount of ferrite, resulting in the reduction of thermoplastic and formation of the low-temperature ductility trough. However, the small difference in thermoplasticity in the low-temperature ductility trough was attributed to the small amount of ferrite and the low transformation temperature of ferrite. The tensile fracture at different temperatures was characterized by means of optical microscopy and scanning electron microscopy. It was found that there were $Al_2O_3$, AlN, MnO, and MnS(Se) impurities in the fracture. The abnormal points of thermoplasticity showed that the inclusions had a significant effect on the high-temperature mechanical properties. The results of EBSD local orientation difference analysis showed that the temperature range with good plasticity was around 1275 °C. Under large deformation extent, the phase difference in the internal position of the grain was larger than that in the grain boundary. The defect density in the grain was large, and the high dislocation density was the main deformation mechanism in the high-temperature tensile process.

**Keywords:** TWIP/TRIP steel; high-temperature mechanical properties; reduction of area; ductility trough



## 1. Introduction

Fe–Mn–C–Al TWIP/TRIP steel has great potential in automotive structural components, the construction industry, and oil and gas exploration due to its excellent tensile strength, ductility, and high energy absorption capacity [1]. In modern automotive structural parts, excellent stamping ductility is expected. This steel provides high-strength structural performance and can absorb a lot of energy to enhance crashworthiness, which is very important for vehicles. In addition, large-scale use of such high-grade steel can reduce vehicle density, reduce the weight of the vehicle itself, reduce greenhouse gas emissions, increase gasoline mileage, and improve passenger safety [2]. The good mechanical properties required can be achieved by strain-induced martensitic transformation or twin materials, which are called TRIP (transformation-induced plasticity) steel and TWIP (twinning-induced plasticity) steel [1]. The difference in the deformation mechanism

between the two steels is caused by stacking fault energy (SFE) of the austenite matrix. The main factors affecting SFE are alloy composition and deformation temperature [3]. Jin et al. [4] studied the effect of Al content in the range of 0–2% on SFE of TWIP steel with Fe–18Mn–0.6C–xAl composition. The results showed that the SFE increased linearly with a constant slope of 7.8 mJ/m$^2$ for every 1 wt % Al addition. Frommeyer et al. [5] showed that TRIP was the main effect when the manganese content was lower than 20%, while the TWIP became the main effect when the manganese content was higher than 25%. Lee et al. [6] studied the tensile properties of medium manganese steel with Mn% in the range of 9–12%. They found that the plastic deformation mechanism was related to the strain rate, and the TRIP and TWIP effects occurred simultaneously when the strain rate was in the range of $10^{-4}$ s$^{-1}$ to $10^{-2}$ s$^{-1}$. However, there have been few studies on the thermoplasticity of steel with TWIP/TRIP effects, which will affect the surface cracking of steel billets during the hot rolling and continuous casting processes. Gennari et al. [7] carried out an innovative heat treatment (intercritical annealing at 780 °C and austempering at 400 °C for 30 min) for a new high-silicon steel with TRIP effect. Higher hardness and higher tensile strength were obtained by isothermal quenching. The high amount of martensite was responsible for the low fracture strain and ductility. Improvement of austenite stability increased the ultimate tensile strength and total elongation of TRIP steel [8]. Furthermore, Mou et al. [9] found that lamellar austenite was more likely to cause stress relaxation during martensitic transformation, resulting in a discontinuous TRIP effect and thus higher ductility. Peng et al. [10] carried out extension experiment at elevated temperature on Fe–22Mn–0.7C TWIP steel in the temperature range of 700–1250 °C and found that the reduction of area was extremely low, all below 40%. Barbieri et al. [11] carried out tensile tests of Fe–22Mn–0.65C TWIP steel at 25–400 °C. Metallographic observation showed that mechanical twins were the main feature of the microstructure at 25 °C. At 350 °C, twins were observed only in some dispersed grains, with dislocation bands observed. Koyama et al. [12] tested tensile Fe–18Mn–1.2C TWIP steel in the temperature range of −150 to 250 °C and analyzed the twin density at different temperatures by EBSD. It was found that the twin density decreased monotonously with the increase in deformation temperature from −50 to 200 °C and became nearly zero at 200 °C. It is well known that deformation-induced ε-martensitic transformation occurs at a lower temperature range than deformation twins, both of which are suppressed with increasing SFE.

The purpose of this study was to examine the mechanical properties and fracture mechanism of TWIP/TRIP steel at temperatures of 600–1310 °C, which is a more representative range. The tensile test was carried out by a thermal simulator at high temperature, and the fracture behavior, phase transformation, and dynamic recrystallization (DRX) were characterized by optical microscopy (OM), scanning electron microscopy (SEM), energy-dispersive spectrometry (EDS), electron backscattered diffraction (EBSD), and the Thermo-Calc thermodynamic software. The high-temperature mechanical properties of Fe–Mn–C–Al TWIP/TRIP were analyzed. The change of SFE in the temperature range of 600–1310 °C was calculated by a formula. The reduction of area (RA) was used to quantify the thermoplasticity, which was compared with Fe–24.2Mn–3Al–2.6Si TWIP steel and 12Cr1MoVG steel. The TWIP/TRIP steel studied here had excellent thermoplasticity in the whole tensile temperature range. There was no traditional ductility trough (≤40%). The high- and low-temperature ductility troughs shifted to the high- and low-temperature zones, respectively, resulting in the excellent performance of the steel in the whole high-temperature zone. This is a very interesting phenomenon. The fracture mechanism and high-temperature mechanical properties are described in detail below.

## 2. Sample Processing and Experimental Methods

### 2.1. Sample Preparation and Sampling

First, 20 kg of Fe–15.3Mn–0.58C–2.3Al TWIP/TRIP steel ingot was produced in a 25 kg medium frequency vacuum induction furnace (Santai Electric Furnace Factory, Jinzhou, China). After removing the riser, the section size was Φ120 mm. The edge of the billet

was selected along the direction shown by the red arrow in Figure 1. A total of 14 tensile specimens with the size of Φ10 × 121.5 mm were selected by wire cutting. The length of the two ends of the thread was 15.25 mm, and the thread spacing was 1.5 mm.

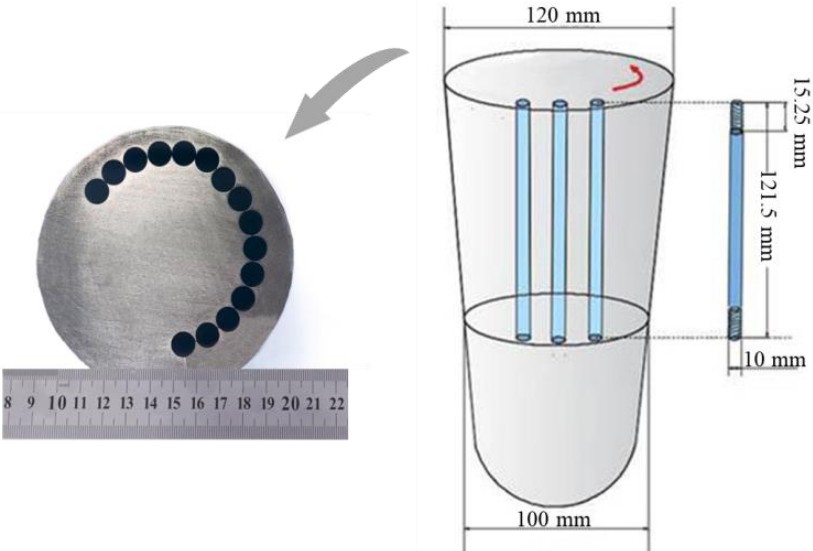

**Figure 1.** Sampling diagram of high-temperature tensile sample.

The mass fraction of the main elements in this steel is shown in Table 1. The concentrations of carbon and sulfur were determined by the infrared absorption method after combustion in oxygen using a CS-230 high-frequency infrared carbon sulfur analyzer (Yanrui Instrument Co., Ltd., Chongqing, China). In addition, the concentrations of nitrogen and oxygen were measured by the thermal conductivity and infrared absorption methods using a NO-330 nitrogen oxygen analyzer (Yanrui Instrument Co., Ltd., Chongqing, China) and helium as carrier gas. The contents of other alloying elements (Mn and Al) were determined by X-ray fluorescence (XRF) (Shimadzu Enterprise Management (China) Co., Ltd., Shanghai, China) under vacuum.

**Table 1.** Chemical composition of specimens (wt %).

| Element | Fe | Mn | C | Al | N | O | S |
|---------|-----|------|------|-----|--------|--------|--------|
| Content (%) | Bal. | 15.3 | 0.58 | 2.3 | 0.0057 | 0.0049 | 0.0099 |

*2.2. High-Temperature Tensile Test*

The processed tensile sample bar was subjected to high-temperature tensile test using a Gleeble-3500 thermal simulator (Fuller Instrument Technology (Shanghai) Co., Ltd, Shanghai, China), and the tensile temperature range was 600–1310 °C. The tensile sample was horizontally fixed in the vacuum chamber. After vacuum pumping, the sample was heated by a resistor in an Ar-filled atmosphere to prevent oxidation. The effective length of heating was 20 mm. Temperature was measured using an R-type platinum–rhodium thermocouple wire. The processing flow chart of the high-temperature tensile specimen is shown in Figure 2. The 12 tensile specimens were heated at room temperature at a rate of 10 °C/s to 1250 °C for 180 s. The purpose of this operation was to carry out solid solution treatment and simulate the conditions appearing in the straightening operation process of continuous casting as much as possible. It was ensured that the steel was fully austenitized, internal stress was eliminated, and a larger grain size was obtained, similar to the grain size of the continuous casting process. Then, it was heated or cooled at a rate of 3 °C/s to predetermined test temperatures (600, 700, 800, 900, 1000, 1100, 1200, 1250, 1275, 1285, 1300, and 1310 °C). After holding for 60 s, the specimens were stretched to

fracture failure at a strain rate of $1 \times 10^{-3}$ s$^{-1}$. Quick cooling was immediately carried out to retain the high-temperature fracture morphology and microstructure. During the high-temperature tensile process, the stress and strain values of the specimen were recorded in real time by a Gleeble-3500 thermal simulator. The curves processed by the Origin software (2021, OriginLab, Hampton, USA). The maximum tensile stress in the tensile process at each temperature was obtained from the recorded data and then compared with Fe–24.2Mn–3Al–2.6Si TWIP steel [13] and 12Cr1MoVG steel [14].

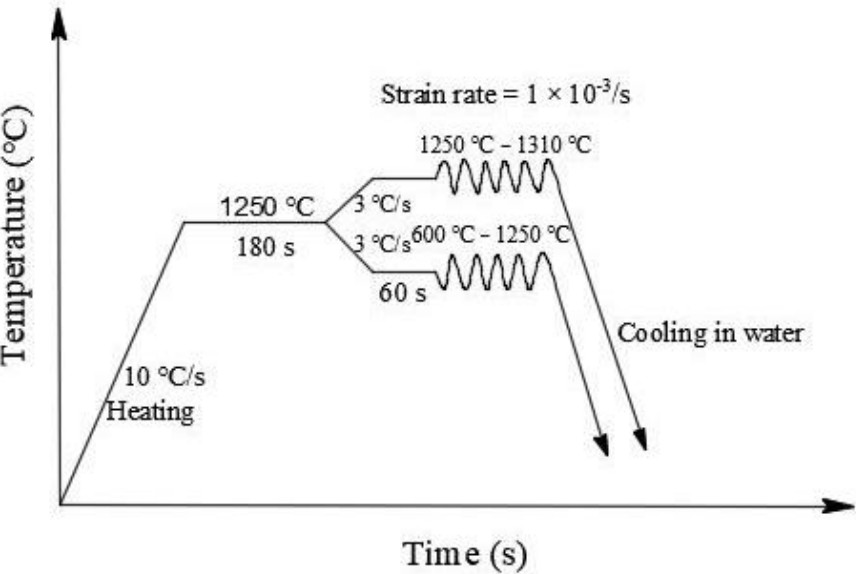

**Figure 2.** Flowchart of high-temperature tensile specimen processing.

As shown in Figure 3, the end with relatively complete morphology after fracture was selected, the fracture was sealed, and samples of appropriate size were cut by wire. Quanta FEG 250 (FEI, Hillsboro, OR, USA) and Zeiss-Utra55 (Zeiss, Oberkochen, German) field emission scanning electron microscopes were used for SEM and EDS tests, respectively. SEM was used to observe the morphology of typical inclusions near the fracture. EDS was used to analyze the composition of the inclusions and the mass fraction of each element. The fracture was cut longitudinally along the tensile direction. After a series of standard grinding and polishing processes, the metallographic surface was etched by 4% nitric acid alcohol solution. The microstructure near the fracture was observed by OLYMPUSGX71 optical microscope (Olympus, Tokyo, Japan). Oxford Nordlys Max3 (Oxford, UK) EBSD was used to detect and analyze the section. Finally, Thermo-Calc thermodynamic software (TCFEC7, Stockholm, Sweden) of TCFE7 database was used to predict the phase change behavior in the whole tensile temperature range.

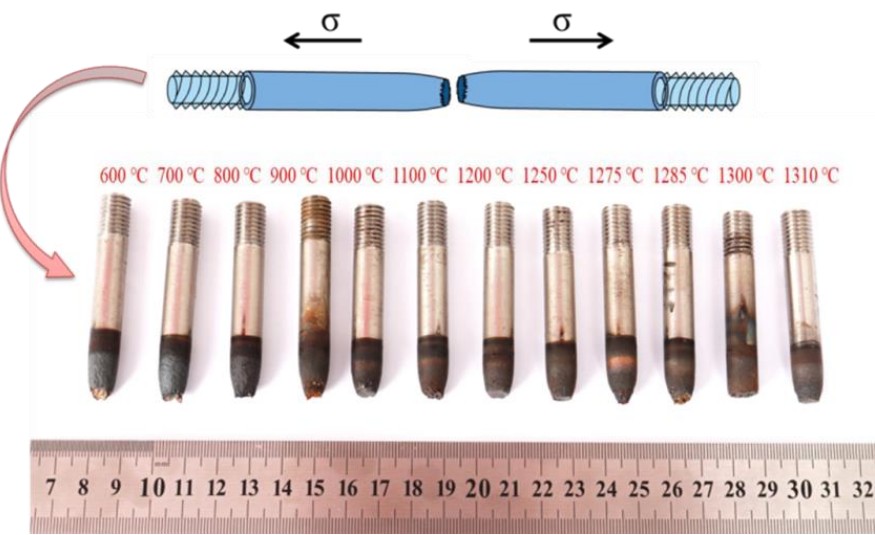

**Figure 3.** Fracture topography.

## 3. Results and Analysis

### 3.1. Calculation of SFE

In Olson's work [15,16], the thermodynamic model of stacking fault energy is expressed as follows:

$$\gamma_{SF} = 2\rho_A \Delta G^{\gamma \to \varepsilon} \tag{1}$$

In Equation (1), $\rho_A$ is the atomic surface stacking density in the face-centered cubic phase, and $\Delta G^{\gamma \to \varepsilon}$ is the free energy difference between the face-centered cubic and close-packed hexagonal phases, which is the key to calculating the stacking fault energy. Allain et al. [17] gave the average value of 10 mJ/m$^2$ for the range of 5–15 mJ/m$^2$.

$$\rho_A = \frac{4}{\sqrt{3}} \cdot \frac{1}{\alpha^2 N} \tag{2}$$

In Equation (2), N is the Avogadro constant and $N = 6.02 \times 10^{23}$, and $\alpha$ is the lattice constant of the alloy. According to the work done in [18,19], the empirical formula of the lattice parameters of Fe–Mn–Al–C alloy and the lattice constant of austenite at any transformation temperature can be derived.

$$\alpha_{\gamma 0} = 0.35721 + 0.0001166 \,(Mn) + 0.0044089 \,(C) + 0.0002771(Al) \tag{3}$$

$$\alpha_{\gamma} = \alpha_{\gamma 0}[1 + \beta_{\gamma}(T - 300)] \tag{4}$$

In Equation (3), the brackets denote the mass fraction of the element. In Formula (4), $\beta\gamma$ denotes the linear thermal expansion coefficient of austenite, $\beta\gamma = 2.065 \times 10^{-5} \,K^{-1}$. $\alpha_{\gamma 0}$ is the lattice parameter of austenite at room temperature, T is Kelvin temperature.

$$\Delta G^{\gamma \to \varepsilon} = \sum X_x \Delta G_x^{\gamma \to \varepsilon} + \sum X_x X_{Fe} \Delta G_{Fex}^{\gamma \to \varepsilon} + X_C \Delta G_{FeMnAl/C}^{\gamma \to \varepsilon} + \Delta G_{mg}^{\gamma \to \varepsilon} \tag{5}$$

In Equation (5), $\Delta G_x^{\gamma \to \varepsilon}$ is the free energy difference between the face-centered cubic and close-packed hexagonal phases of Fe, Mn, Al, and C alloy elements. $X_x$ is the molar fraction of Fe, Mn, Al, and C alloy elements, and $\Delta G_{Fex}^{\gamma \to \varepsilon}$ is the interaction parameter difference between Fe and Mn, Al, and C alloy elements. Here, only the interaction between Fe and Mn is considered, and the interaction between iron and other alloy elements is

ignored. $\Delta G^{\gamma\to\varepsilon}_{FeMnAl/C}$ is the effect of interaction between Fe, Mn, Al, and C on phase transition free energy.

$$\Delta G^{\gamma\to\varepsilon}_{FeMnAl/C} = \frac{1246}{X_C}\left[1 - \exp(-24.29\,X_C)\right] - 17175X_{Mn} \tag{6}$$

$\Delta G^{\gamma\to\varepsilon}_{mg}$ is the difference of molar magnetic free energy between the $\gamma$ and $\varepsilon$ phases.

$$\Delta G^{\gamma\to\varepsilon}_{mg} = G^{\varepsilon}_m - G^{\gamma}_m \tag{7}$$

In Equation (7), $G^{\varepsilon}_m$ and $G^{\gamma}_m$ represent the molar magnetic free energy of the $\varepsilon$ and $\gamma$ phases, respectively.

$$G^{\varepsilon}_m = RT\ln\left(\beta^{\varepsilon}+1\right)f^{\varepsilon}(\tau^{\varepsilon}), \quad G^{\gamma}_m = RT\ln\left(\beta^{\gamma}+1\right)f^{\gamma}(\tau^{\gamma}) \tag{8}$$

$$f(\tau) = -\frac{1}{D}\left(\frac{\tau^{-5}}{10} + \frac{\tau^{-15}}{315} + \frac{\tau^{-25}}{1500}\right) \tag{9}$$

$$\tau^{\varepsilon} = \frac{T}{T^{\varepsilon}_N}, \quad \tau^{\gamma} = \frac{T}{T^{\gamma}_N} \tag{10}$$

In Equation (10), T is the actual temperature, and $T_N$ is the temperature of the antiferromagnetic transition point.

$$T^{\gamma}_N = 250\ln X_{Mn} - 4750X_{Mn}X_C - 6.2X_{Al} + 720\,(k) \tag{11}$$

$$T^{\varepsilon}_N = 580X_{Mn}\,(k) \tag{12}$$

$$\beta^{\gamma} = 0.7X_{Fe} + 0.62X_{Mn} + 0.64X_{Fe}X_{Mn} - 4X_C \tag{13}$$

$$\beta^{\varepsilon} = 0.62X_{Mn} - 4X_C \tag{14}$$

$$D = \frac{518}{1125} + \frac{11692}{15975}\left(\frac{1}{P} - 1\right) \tag{15}$$

For fcc and hcp crystal structures, P = 0.28 in Equation (15). In addition, the physical parameters involved are shown in Table 2.

**Table 2.** Relevant physical parameters required to calculate SFE.

| Physical Parameter [20] | Value/(J·mol$^{-1}$) |
|---|---|
| $\Delta G^{\gamma\to\varepsilon}_{Fe}$ | $-2243.38 + 4.309T$ |
| $\Delta G^{\gamma\to\varepsilon}_{Mn}$ | $-1000.00 + 1.123T$ |
| $\Delta G^{\gamma\to\varepsilon}_{C}$ | $-22,166$ |
| $\Delta G^{\gamma\to\varepsilon}_{Al}$ | $2800 + 5T$ |
| $\Delta G^{\gamma\to\varepsilon}_{FeMn}$ | $2873 - 717(X_{Fe} - X_{Mn})$ |
| $\Delta G^{\gamma\to\varepsilon}_{FeC}$ | $42,500$ |
| $\Delta G^{\gamma\to\varepsilon}_{FeAl}$ | $3339$ |

Through the above formula and related physical parameters, the SFE value of this steel at 25 °C was 15.6 mJ/m$^2$, and the SFE value range at the tensile temperature of 600–1310 °C was 120.8–221 mJ/m$^2$. When the composition of TRIP/TWIP steel was unchanged, the SFE value increased with the increase in temperature, which is consistent with the conclusions of Akbari [21] and Allain et al. [17]. In addition, martensitic transformation occurred when SFE was <18 mJ/m$^2$, which is beneficial to martensite-induced plasticity. When the SFE value was 18–45 mJ/m$^2$, the martensitic transformation was inhibited, and the mechanical twinning was enhanced. When the SFE value is ≥45 mJ/m$^2$, the dislocation slip is a single deformation mechanism [15]. Combined with the above, the TRIP effect was the main deformation mechanism of steel at room temperature. The dislocation slip mechanism was dominant between 600 and 1310 °C.

### 3.2. Tensile Strength Analysis

The effect of deformation temperature on the tensile properties of Fe–Mn–C–Al TWIP/TRIP steel is shown in Figure 4. Figure 4a is the true stress–strain curve in the temperature range of 600–1310 °C. As can be seen from the curve, there was a single peak stress from 600 to 800 °C, and the maximum stress was about 395 Mpa at 600 °C. The stress decreased sharply after reaching the peak value, and the specimen was fractured. Moreover, the decreasing trend slowed down with the increase in temperature. As can be seen from the curve, there was fluctuation at 900 °C and a short processing rigidification area on the true stress–strain curve of the subsequent temperature. The stress increased rapidly with the increase in strain and then reached the maximum tensile stress, which is the ultimate tensile strength [22]. Then, a long post ultimate tensile strength was formed, which might be related to the occurrence of DRX [23]. Figure 4b compares the tensile strength of Fe–24.2Mn–3Al–2.6Si TWIP steel and 12Cr1MoVG steel with that of the steel tested in this study. It can be seen that the peak stress of the three steels decreased with the increase in tensile temperature. In the range of 600–850 °C, the tensile strength of this steel was between the other two steels. However, above 850 °C, the tensile strength became relatively excellent. This proved that Fe–15.3Mn–0.58C–2.3Al TWIP/TRIP steel had better tensile strength than the two other steels at high temperature.

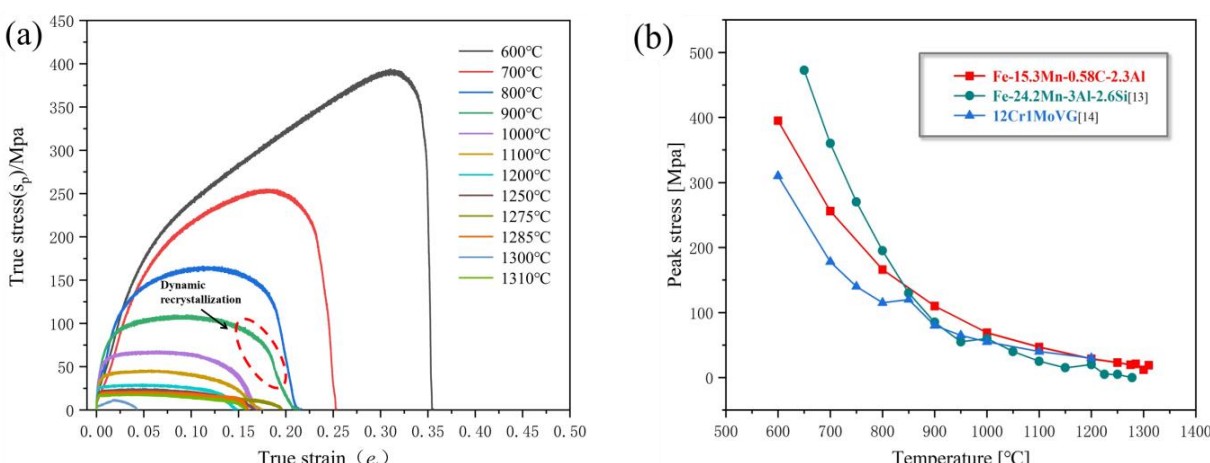

**Figure 4.** (**a**) True stress–strain curve and (**b**) curve of peak stress versus temperature.

### 3.3. Thermoplastic Analysis

Elongation at break complicates the interpretation of the curve due to necking. Thermoplastic is quantified as the percentage reduction in the cross-sectional area of the sample at fracture. The cross-sectional area after fracture was measured by standard vernier caliper, and the RA of Fe–15.3Mn–0.58C–2.3Al TWIP/TRIP steel was calculated using Equation (16). The variation trend of the RA of this steel with temperature is shown by the red curve in Figure 5. In addition, for comparison, the RA of Fe–24.2Mn–3Al–2.6Si and 12Cr1MoVG were compared.

$$\varphi = \frac{A_0 - A_f}{A_0} \times 100\% \tag{16}$$

In Equation (16), $A_0$ and $A_f$ are the cross-sectional areas of the specimen before and after fracture, respectively.

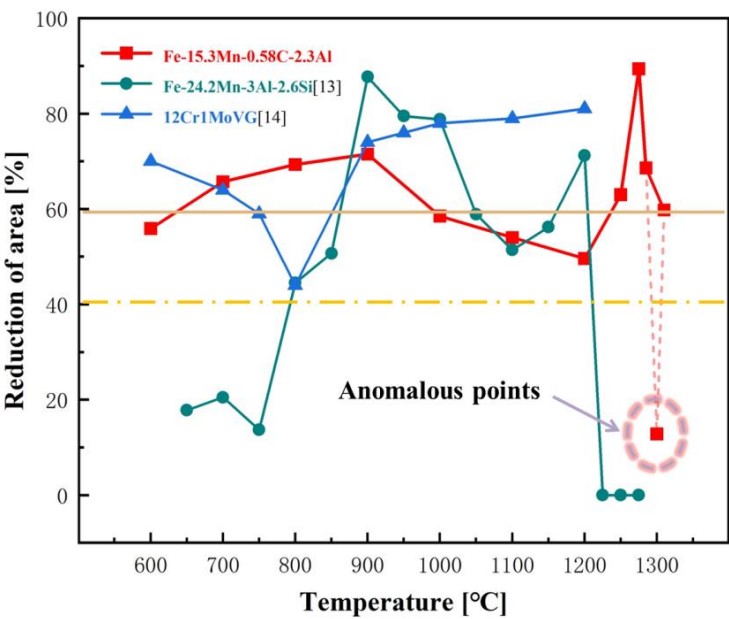

**Figure 5.** Temperature dependence of reduction of area of Fe–15.3Mn–5.8C–2.3Al TWIP/TRIP steel and Fe–24.2Mn–3Al–2.6Si and 12Cr1MoVG steel.

According to the thermoplastic curve of Fe–15.3Mn–0.58C–2.3Al TWIP/TRIP steel, with the increase in temperature, the plasticity increased in the temperature range of 600–900 °C, and the RA was 71.5% at 900 °C. Between 1000 and 1250 °C, the medium temperature ductility trough of this steel had the worst plasticity in the whole temperature range at 1200 °C, with RA of 47.3%. Because this experiment was carried out at a low strain rate ($1 \times 10^{-3}$ s$^{-1}$), the embrittlement would not be serious. As the temperature continued to rise, the plasticity increased rapidly. At 1275 °C, the RA reached the maximum, increasing to 89.4%. The high-temperature ductility trough was from 1275 °C to the melting point. However, with the increase in temperature of only 25 °C, i.e., at 1300 °C, the thermoplastic curve appeared as an anomalous point. Plasticity suddenly deteriorated sharply, and a brittle fracture occurred; the RA was only 12.8%. Through in-depth analysis of the anomalous point, it was found that there was a large number of Al$_2$O$_3$, AlN, MnO, and MnS(Se) inclusions at the fracture of the sample. According to the thermodynamic calculation and experimental study in [24], it was found that the AlN precipitated in the liquid phase, and as the heterogeneous nucleation core, MnS(Se) and other inclusions precipitated locally on its surface. The aggregation of brittle inclusions seriously deteriorated the high-temperature mechanical properties, which was the fundamental reason for the significant reduction in RA. The fracture morphology observation in Figure 3 shows that there was no obvious necking phenomenon in the tensile specimen at this temperature. In Figure 5, the RA of Fe–15.3Mn–0.58C–2.3Al TWIP/TRIP steel in this study is compared with that of Fe–24.2Mn–3Al–2.6Si TWIP steel in Li et al. [13] and 12Cr1MoVG steel in Dong et al. [14] at high temperature. As can be seen, the TWIP/TRIP steel in this study had better plasticity at high temperature. In the temperature range of 600–1310 °C, the RA was 47.3–89.4%. According to the experimental study of several steel grades in the literature, 40% RA is the critical value of ductility trough [25]. When the reduction of area is less than 40%, cracks will increase. No traditional ductility trough in the temperature range of 600–1310 °C has been measured in this steel. In this study, the relative ductility trough was below 60% RA. The RA of Fe–24.2Mn–3Al–2.6Si TWIP steel and 12Cr1MoVG steel in the same temperature range were 0–90% and 40–82%, respectively. There was an obvious ductility trough below 850 °C.

Research has shown that it is easy to generate holes and cracks on the ferrite precipitated on the austenite grain boundary, which is the possible reason for intergranular fracture. The Thermo-Calc thermodynamic calculation of this steel (see Section 3.6) showed

a small amount of ferrite was precipitated on austenite between 600 and 700 °C, resulting in the decrease in RA from 65.7 to 55.9%. However, ferrite was found in Fe–24.2Mn–3Al–2.6Si TWIP steel at 850 °C. Therefore, the decrease in ferrite transformation temperature caused the low-temperature ductility trough to shift to the low-temperature section. The small amount of ferrite precipitation meant there was only a slight effect on the RA. In addition, it was found that the high-temperature ductility trough of the steel in this study was obviously shifted to the high-temperature zone compared to the other two steels. Zero ductility temperature (ZDT) and zero strength temperature (ZST) were estimated at about 1330 °C. In summary, the Fe–15.3Mn–0.58C–2.3Al TWIP/TRIP steel had comparatively good plasticity in the whole tensile temperature range.

### 3.4. Fracture Morphology

The fracture of Fe–15.3Mn–0.58C–2.3Al TWIP/TRIP steel at different temperatures was observed by Quanta FEG 250 and Zeiss-Utra55 field emission scanning electron microscopy, as shown in Figure 6. The red box in the lower left corner is a local zoom of the area referred to by the blue arrow. When the test temperature was 600 °C, as shown in Figure 6a, there were a large number of small-sized circular or elliptical dimples, but large-sized dimples, which belong to intergranular dimple fracture, were rare. With the increase in temperature, the dimple size increased, and some large and deep equiaxed dimples appeared. In addition, there were many small dimples around the large dimples. As can be seen from Figure 6b,c, with smaller inclusions, there were second-phase particles or micropores between the large dimples [26]. As plastic deformation continued, the connection between large dimples could be carried out through secondary dimples. $MnS(Se) + MnO + Al_2O_3 + AlN$ composite inclusions were observed near the wedge cracks. Figure 6d clearly shows the morphology characteristics of the above small dimples and the MnS (Se) + MnO inclusions in the concave dimples. There were many typical dimples with small size, shallow depth, and tearing ridge. In addition, there were some large dimples with a diameter of about 400 μm at 900 °C. As the dimple diameter increased, the deeper the depth, the better was the plasticity. This was consistent with the phenomenon of the RA increasing from 55.9 to 71.5% in the temperature range of 600–900 °C, as shown in Figure 5. From Figure 6f, it can be observed that the number of dimples decreased at 1100 °C. The fracture was smooth, there was no characteristic small plane, and a large number of wedge cracks appeared. This is the intergranular debonding phenomenon caused by slip due to the weakening of grain boundary strength during the tensile process [27,28], which belongs to intergranular brittle fracture. As Mejía et al. [29] reported, local cracking is a natural result of inhomogeneous plastic deformation. At the temperature of 1200 °C, no dimples were found in the whole fracture field. As shown in Figure 6g, the structure between the dendrites was in a layered state after tearing and breaking. With the increase in temperature, the plasticity reached the best value at 1275 °C, as shown in Figure 6h,i, and the fracture was cup–cone. Strong plastic deformation occurred in the unfractured area, which gradually formed a cavity. MnS (Se) inclusions were observed near the dimples. In addition, it was also found that there was a hunting slip pattern on the inner wall of the larger dimple. This is because when the dimple surface is perpendicular to the principal stress direction, new slip occurs on the surface of the dimple under the action of stress. The primary slip trace was sharp. With the continuous slip, it developed smoothly into a hunting pattern and further flattened, which is a typical ductile fracture feature.

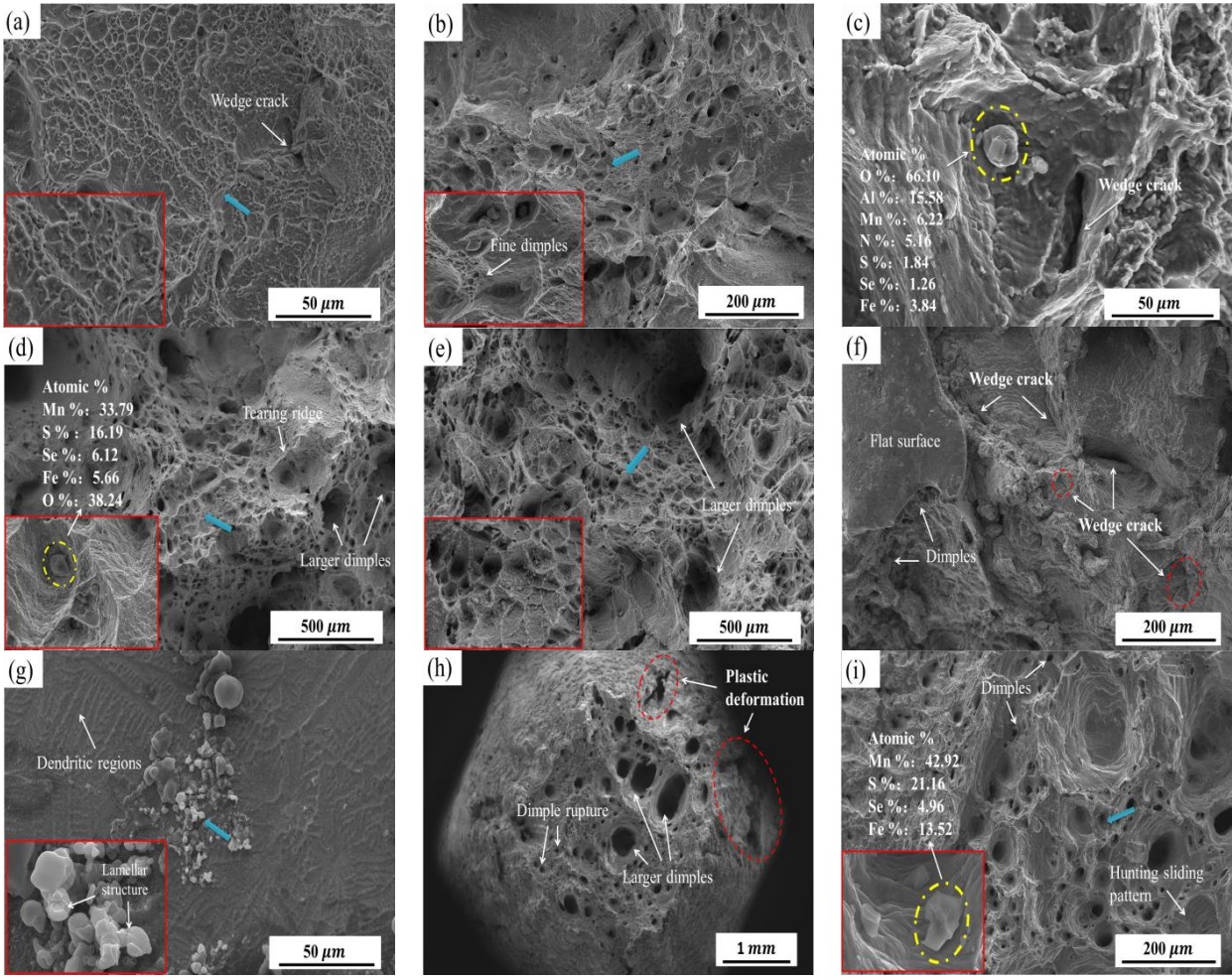

**Figure 6.** Fracture morphology of samples at different test temperatures. (**a**) 600 °C; (**b**,**c**) 700 °C; (**d**) 800 °C; (**e**) 900 °C; (**f**) 1100 °C; (**g**) 1200 °C; (**h**,**i**) 1275 °C.

### 3.5. Microstructure

Figure 7 shows the microstructure of the samples with different tensile temperatures and without tensile at room temperature. The stretching direction has been marked by a white arrow. Figure 7a is the microstructure of the sample at 25 °C, where a large number of isometric austenite grains can be observed. Figure 7b–f is a picture of the microstructure around the fracture at different tensile temperatures. Figure 7c is a microstructure about 5 mm away from the fracture at 700 °C. Compared with Figure 7b, it can be observed that the number of cracks and voids at the trifurcate grain boundary was significantly reduced, and the austenite grain was not elongated. Large cavities were initially formed at the early stage of deformation due to grain boundary sliding. Grain boundary sliding is the main fracture mechanism at this temperature. Then, when the crack propagates, it distorts into slender cavity until it finally fails [23,30]. Dislocation pile-up can be easily formed at the triple line boundary, resulting in stress concentration, and tensile microcracks can be easily generated. With the extension, the microcracks continue to expand and connect with each other, resulting in the fracture of the sample. At 1200 and 1250 °C, an obvious dendritic structure was observed near the fracture. Combined with the observation in Figure 6f,g, it was found that in this temperature range, some fracture surfaces had the re-melting phenomenon. This might have been due to the segregation of P, S, and other elements on the grain boundary, which reduced the melting point of the grain boundary and was the main factor leading to the occurrence of a dendritic structure. The thermoplasticity in this temperature range was reduced, and the RA, shown in Figure 5, was also significantly

reduced. A large number of intergranular cracks and micropores were formed near the fracture surface and existed between dendrites, which meant that cracks could easily occur and propagate along the dendrite boundary. When the micropores are serious enough, the cracks can quickly connect and fracture. Especially at 1200 °C, there were a large number of micropores, which led to the occurrence of brittle fracture and a decrease of section shrinkage. When the temperature reached 1275 °C, there were subgrain boundaries in some slender grains near the fracture front, which might be partially related to dynamic recovery and dynamic recrystallization. The decrease in grain size could be attributed to continuous DRX caused by dislocation accumulation [31]. Some grain sizes increase because the essence of thermally activated process is thermal activation [32]. Under high temperature tensile process and low strain rate, the ability of thermal activation is strong. It is easy to overcome energy barriers, and grain boundary migration can easily occur, which promotes the nucleation and growth of DRX [33]. This is also the main reason for the high section shrinkage in this temperature range.

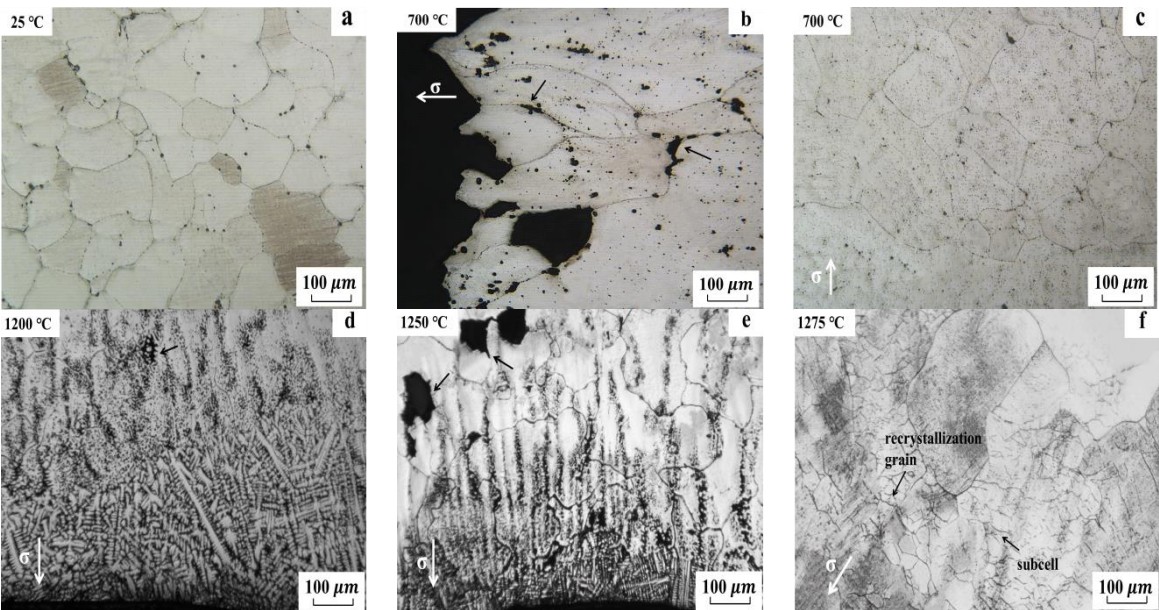

**Figure 7.** Microstructure morphology near the high-temperature tensile fracture. (**a**) 25 °C; (**b,c**) 700 °C; (**d**) 1200 °C; (**e**) 1250 °C; (**f**) 1275 °C.

### 3.6. Phase Transition Process and EBSD Analysis

The phase change process of the steel was calculated by Thermo-Calc thermodynamic software. It should be noted that in order to be rigorous, sulfur, oxygen, and nitrogen elements with relatively low content should be considered in the calculation process. According to the thermodynamic calculation, as shown in Figure 8a, there was no single austenite structure in the whole tensile temperature range. The main phases were austenite face-centered cubic phase, MnS, AlN, and $Al_2O_3$. It can be observed from Figure 8b that austenite began to appear at 1429.3 °C. Among them, AlN (1458.9 °C) and $Al_2O_3$ (2029.8 °C) were formed at a temperature higher than their liquidus, and the solidus was 1343.1 °C. The formation temperature of MnS was 1328.1 °C, which was slightly lower than the solidus. The results showed that AlN and $Al_2O_3$ began to form in liquid high manganese steel. With the solidification process, inclusions gradually increased and eventually tended to be stable. AlN tended to be stable at about 1000 °C with a molar fraction of $4.2 \times 10^{-4}$. As shown in Figure 8a, there was a small amount of ferrite body-centered cubic phase and cementite between 600 and 700 °C, with the cementite generated at 665.3 °C. It disappeared quickly when the temperature dropped to 621.7 °C. Thus, the presence of cementite had no effect on the tensile temperature in the experiment.

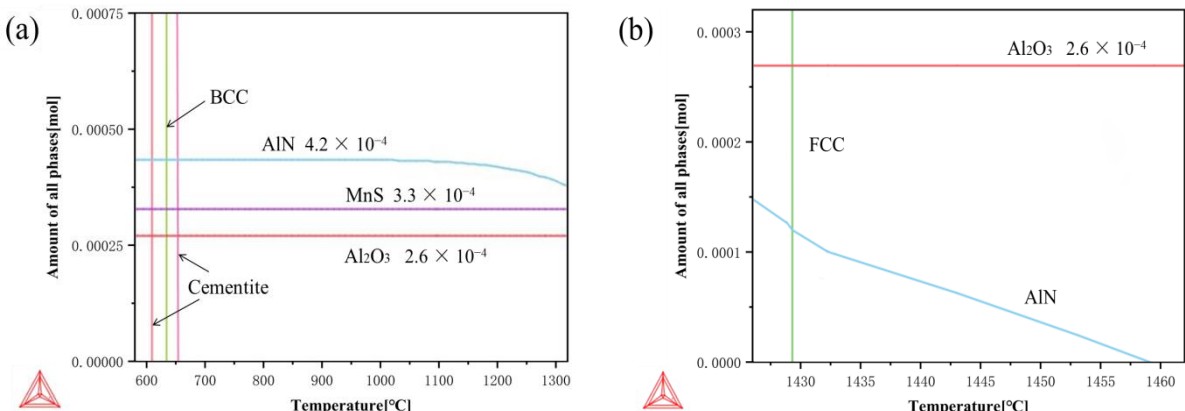

**Figure 8.** Predicted phase change behavior of steel by Thermo-Calc thermodynamic calculation. (**a**) 600–1300 °C; (**b**) 1425–1460 °C.

It was found that large deformation increased dislocation density and increased the driving force of transformation from austenite to ferrite. The thermodynamic calculation, shown in the diagram above, showed that a ferrite body-centered cubic phase appeared at 646 °C and a small amount of ferrite precipitated, which softened the austenite. However, due to the low formation temperature and low content of ferrite, the RA at 600–700 °C was less affected and the curve decreased gently, as shown in Figure 5. Compared with Fe–24Mn–2.6Si–3Al steel, the liquidus temperature of this steel was increased by about 100 °C, which would greatly improve the temperature of the high-temperature ductility trough and delay the embrittlement caused by the liquid film at the crystal interface [34].

Oxford Nordlys Max3 EBSD was used for surface scan, and the scan results were analyzed by local misorientation. The dislocation density is very high in the deformed grains produced by high-temperature stretching. These dislocations are arranged in the dislocation structure, resulting in different degrees of local misorientation [35]. Local misorientation can be used to study the orientation change within grains during plastic deformation so as to measure the relative size of dislocation density in deformed metals. The higher the whole dislocation density, the larger is the average local misorientation [36]. Figure 9a,b shows the grain boundary distribution and local orientation difference distribution as characterized by EBSD at 1200 and 1275 °C. The orientation deviation near the fracture is also shown. In the color scale, yellow-green color represents a large orientational angle. Although the upper threshold of local misorientation was set to 5.0°, the actual local misorientation was mostly concentrated below 2.0°. It can be seen intuitively that at 1200 °C, most areas are blue areas with small local misorientation. Only a small number of green regions with large local misorientation are distributed on the grain boundaries of subgrains. In contrast, at 1275 °C, the yellow-green region accounts for most of the area. It is mostly located in the grain interior, and a small amount is located in the grain boundary. This indicates that when the temperature increased from 1200 to 1275 °C, the grain orientation deviation angle increased compared with the grain boundary. The deformation occurred in most areas of the grain, and the deformation was large. The defect density increased and the internal stress increased, resulting in a large number of dislocations. The dislocation density was high, and the speed of dislocation disappearance was also accelerated. This further indicates that the dislocation slip mechanism was dominant.

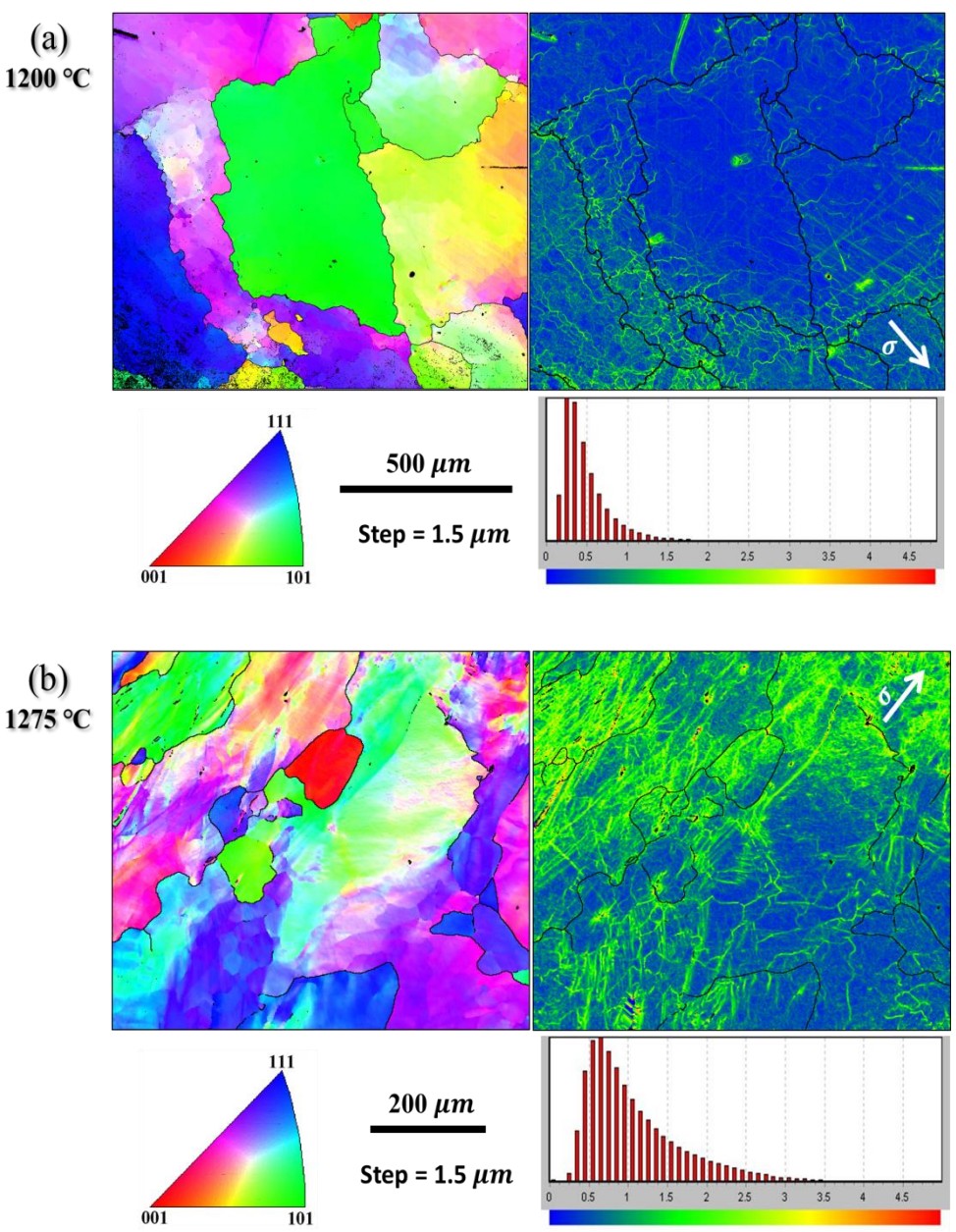

**Figure 9.** EBSD analysis near tensile fracture at 1200 and 1275 °C. (**a**) 1200 °C; (**b**) 1275 °C.

## 4. Conclusions

(1) The Fe–15.3Mn–0.58C–2.3Al TWIP/TRIP steel studied in this work had a good plastic temperature range (600–1310 °C). Compared with the traditional ductility trough (RA of less than 40%), there was no traditional ductility trough in this steel. Taking the RA of 60% as the relative ductility trough, the high-temperature ductility trough (from 1275 °C to the melting point) shifted to the high-temperature range due to the high temperature of the solid line. The medium-temperature ductility trough was between 1000 and 1250 °C. Due to the low precipitation temperature and low content of ferrite, the low-temperature ductility trough shifted to the low-temperature range and existed in the temperature range near 600 °C.

(2) According to the thermodynamic calculation of SFE, the SFE value of this steel was 15.6 mJ/m$^2$ at 25 °C, which belongs to low SFE metal, and it had a TWIP/TRIP effect. In the tested tensile temperature range (600–1310 °C), the SFE value was 120.8–221 mJ/m$^2$. A higher SFE will make the dislocation slip become the dominant mechanism of thermal deformation in the high-temperature tensile process.

(3) When the local misorientation in grains was large, the dislocation density increased and the dislocation disappearance rate accelerated. Moreover, the hindering effect of DRX on crack propagation was the main reason for good plasticity in the temperature range of 1250–1285 °C. Through OM and EBSD observation at 1200 °C, it was found that the rapid connection of many micropores near the fracture under tensile stress was the main factor for the brittle fracture at this temperature. In addition, the aggregation of $Al_2O_3$, AlN, MnO, and MnS (Se) inclusions reduced the thermoplasticity and the RA, which was the fundamental factor affecting the high-temperature mechanical properties of this steel.

**Author Contributions:** Conceptualization, G.Y. and C.Z.; methodology, G.Y., C.Z. and C.L.; software, G.Y., C.Z., F.L. and H.Y.; writing—original draft preparation, G.Y. and C.Z.; writing—review and editing, G.Y. and C.Z.; funding acquisition, C.L. and C.Z. All authors have read and agreed to the published version of the manuscript.

**Funding:** This project is financially supported by the "National Science Foundation of China with the grant numbers 51704083 and 51864013", Guizhou Science and Technology Plan Project ([2017]5788, [2018]1026, [2019]1115, and [2019]2163), Education Department Foundation of Guizhou Province (No. [2017]118), Guizhou Science Cooperation platform talents [2018] 5781 subsidy, and Research Foundation for Talents of Guizhou University (No. 201628).

**Institutional Review Board Statement:** Not applicable.

**Informed Consent Statement:** Not applicable.

**Data Availability Statement:** Not applicable.

**Conflicts of Interest:** The authors declare no conflict of interest.

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
