# Peer review of "Study on High-Temperature Mechanical Properties of Fe–Mn–C–Al TWIP/TRIP Steel"

_metals, doi:10.3390/met11050821_

Round 1
Reviewer 1 Report
The present work is devoted to the study of mechanical properties of a TRIP steel with gleeble testing. The work is interesting and well written hovewer some modifications are required before publication:
a)The abstract is too long, please move part of the abstract in the introduction section
b)Please add and discuss in the introduction section some references regarding other TRIP steels, for example the high silicon ones (for example "Franceschi, M.; Pezzato, L.; Gennari, C.; Fabrizi, A.; Polyakova, M.; Konstantinov, D.; Brunelli, K.; Dabalà, M. Effect of Intercritical Annealing and Austempering on the Microstructure and Mechanical Properties of a High Silicon Manganese Steel. Metals 2020, 10, 1448. https://doi.org/10.3390/met10111448)
c)Please state clearly at the end of the introduction the novelty of the work
d)Please add in the experimental the reproducibility of the results (number of samples tested) and describe how the curves were obtained from the ones of the different samples
e) The microstructure of the initial sample (before high temperature tensile tests) should be useful
f) I suggest to add also the microstructural analysys far from the fracture zone in order to compare it with the one already reported and understand better what happened during the tests
Reviewer 2 Report
- What is a "brittle zone"? If ductility of about 10% is observed, it is not brittle.
- This study does not consider microstructure inhomogeneity in as-cast ingots. The morphology of crystal grains and segregation of elements have a great effect on fracture ductility.
- The authors evaluate ductility based solely on RA. However, for example, the 600°C test material has a relatively small RA, but the fracture elongation shown in Figure 4 is large. Is the ductility evaluation by RA alone appropriate?
- The reference numbers in the manuscript and in the list do not match.
- Please re-check the spell and English grammar of the manuscript. There are some errors in the manuscript. P2 L6 “reduce vehicle quality”, P4 L20 “Avgadro constant”, etc. Please check the accuracy of the description of the manuscript.
- Please describe how to heat the specimen and how to measure the temperature during the high-temperature tensile tests in section 2.2.
- What is the intent of the thermal profile before the tensile test shown in Figure 2? Why did the author not keep the specimen isothermal to the test temperature?
- Please provide references for the description below.
P7 L27 “The research shows that the ferrite precipitated on the austenite grain boundary is easy to generate holes and cracks, which is the main reason for intergranular fracture.”
- Consider why dendrite microstructure is found on the 1200°C and 1250°C fracture surfaces in Figure 7.
- The notation “Metallographic Organization” is not common in metallurgy.
- There is insufficient evidence for the occurrence of recrystallization. The KAM map presented by the author does not support the occurrence of recrystallization.
Round 2
Reviewer 1 Report
Considering that the authors have properly answered to the main issues of the first revision i suggest publication for the present work
Author Response
Thank you for your comments on this manuscript. With your help and the help of the editor, it has become more perfect. I believe it will be helpful to more readers. Thank you again for your time and encouragement!!
Reviewer 2 Report
The authors' responses are inadequate as answers to the reviewers' comments. I would like to add the following supplement to my previous comment.
Point 1
I think it is not common to call a specific temperature range where steel ductility decreases as a "brittle zone". As far as I searched on the internet, I felt that only Chinese researchers call a particular temperature range a "brittle zone". Also, there are some studies that call the low ductility part near the welded HAZ a "brittle zone", so this term is confusing.
Point 3
As the authors explained, fracture elongation is usually closely related to the reduction of area. However, in the case of this study, it is clear that there is no correlation between the two. For example, when the results of this high-temperature tensile test are re-plotted for the relationship between fracture elongation and temperature, the tendency is different from that in Figure 5. Therefore, the authors need to explain the validity of the evaluation of the ductility by RA.
Point 7
I commented on Figure 2. Please explain the purpose of the solution heat treatment at 1250 ℃ -180sec. Also, please explain why it was done as a series of operations with the heating of a tensile test rather than an independent heat treatment.
Point 9
Re-melting during high temperature tensile testing can significantly affect the ductility of the material. If the authors observed it as shown in Figure 7, the authors should discuss the effects of this in the manuscript.
Point 11
The reviewer agrees that DRX can occur during high temperature deformation. However, there is not enough evidence to associate the stress reduction in Figure 4 and the fine grain in Figure 7 with DRX. Rather than Figure 7, the curved grain boundary shape in Figure 9 may indicate that DRX (grain boundary migration) has occurred. However, in order to associate the RA change shown in Figure 5 with the grain boundary migration, it is necessary to compare the grain boundary shape after the tensile test at each temperature.
Round 3
Reviewer 2 Report
The authors have been sufficiently improved their manuscript to warrant publication in Metals.
I thank the authors for explaining the description in the textbook. The phenomenon called "高温脆性区II" in Chinese is called "赤熱脆性" in the reviewer's native language. Through this review, I have reaffirmed the need to be aware of the differences between global scientific terms and simple translations of domestic ones.